# TATA: A Multilingual Table-to-Text Dataset for African Languages

Sebastian Gehrmann[†*]      Sebastian Ruder      Vitaly Nikolaev

Jan A. Botha      Michael Chavinda      Ankur Parikh      Clara Rivera

Google   [†]Bloomberg

## Abstract

Existing data-to-text generation datasets are mostly limited to English. To address this lack of data, we create Table-to-Text in African languages (TATA), the first large multilingual table-to-text dataset with a focus on African languages. We created TATA by transcribing figures and accompanying text in bilingual reports by the Demographic and Health Surveys Program, followed by professional translation to make the dataset fully parallel. TATA includes 8,700 examples in nine languages including four African languages (Hausa, Igbo, Swahili, and Yorùbá) and a zero-shot test language (Russian). We additionally release screenshots of the original figures for future research on multilingual multi-modal approaches. Through an in-depth human evaluation, we show that TATA is challenging for current models and that less than half the outputs from an mT5-XXL-based model are understandable and attributable to the source data. We further demonstrate that existing metrics perform poorly for TATA and introduce learned metrics that achieve a high correlation with human judgments. Our results highlight a) the need for validating metrics; and b) the importance of domain-specific metrics.[1]

## 1 Introduction

Generating text based on structured data is a classic natural language generation (NLG) problem that still poses significant challenges to current models. Despite the recent increase in work focusing on creating multilingual and cross-lingual resources for NLP (Nekoto et al., 2020; Ponti et al., 2020; Ruder et al., 2021), data-to-text datasets are mostly limited to English and a small number of other languages. Data-to-text generation presents important opportunities in multilingual settings, e.g., the expansion of widely used knowledge sources, such

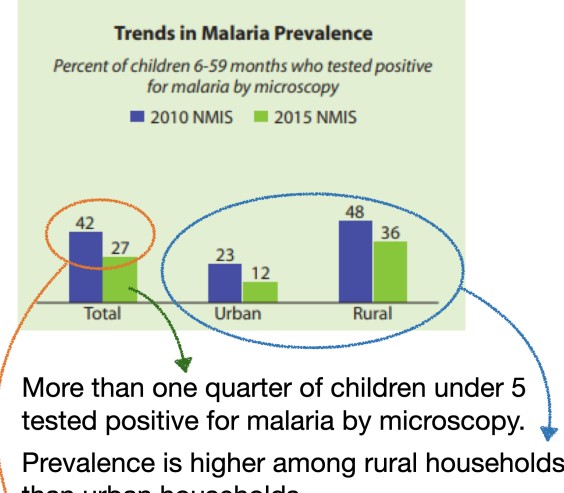

Figure 1: An example from TATA, which demonstrates many of the reasoning challenges it poses.

as Wikipedia to under-represented languages (Lebret et al., 2016). Data-to-text tasks are also an effective testbed to assess reasoning capabilities of models (Suadaa et al., 2021).

However, creating challenging, high-quality datasets for NLG is difficult. Datasets frequently suffer from outputs that are not attributable to the inputs or are unnatural, and overly simple tasks fail to identify model limitations (Parikh et al., 2020; Thomson et al., 2020; Yuan et al., 2021). To provide a high-quality dataset for multilingual data-to-text generation, we introduce Table-to-Text in African languages (TATA). TATA contains multiple references for each example, which require selecting important content, reasoning over multiple cells, and realizing it in the respective language (see Fig. 1). The dataset is parallel and covers nine languages, eight of which are spoken in Africa: Arabic, English, French, Hausa, Igbo, Portuguese, Swahili, Yorùbá, and Russian.[2] We create TATA

---

[*]Work done while at Google.

[1]We release all data at https://github.com/google-research/url-nlp.

[2]The languages were selected based on the availability of

by transcribing[3] and translating charts and their descriptions in informational reports by the Demographic and Health Surveys (DHS) Program, which publishes population, health, and nutrition data through more than 400 surveys in over 90 countries in PDF format.

In an analysis of TATA using professional annotators, we find that over 75% of collected sentences require reasoning over and comparing multiple cells, which makes the dataset challenging for current models. Even our best baseline model generates attributable language less than half of the time, i.e., over half of model outputs are not faithful to the source. Moreover, we demonstrate that popular automatic metrics achieve very low correlations with human judgments and are thus unreliable. To mitigate this issue, we train our own metrics on human annotations, which we call STATA, and we use them to investigate the cross-lingual transfer properties of monolingually trained models. This setup identifies Swahili as the best transfer language whereas traditional metrics would have falsely indicated other languages.

Overall, our experiments highlight that a) metrics need to always be validated in how they are used; and b) that domain-specific metrics may be necessary. Consequently, as our field increasingly relies on trained metrics, e.g., reward models for RLHF (Christiano et al., 2017), we must ensure that metrics are appropriate for the chosen task.

## 2 Background and Related Work

**Data-to-Text Generation** Generating natural language grounded in structured (tabular) data is an NLG problem with a long history (Reiter and Dale, 1997). The setup has many applications ranging from virtual assistants (Arun et al., 2020; Mehri et al., 2022) to the generation of news articles (Washington Post, 2020) or weather reports (Sripada et al., 2004). To study the problem in academic settings, there are two commonly investigated tasks: (1) generate a (short) text that uses all and only the information provided in the input;

(2) generate a description of only (but not all) the information in the input. Corpora targeting the first typically have short inputs, for example key-value attributes describing a restaurant (Novikova et al., 2017) or subject-verb-predicate triples (Gardent et al., 2017a,b). Datasets in the second category include ones with the goal to generate Wikipedia texts (Lebret et al., 2016; Parikh et al., 2020) and sport commentary (Wiseman et al., 2017; Thomson et al., 2020; Puduppully and Lapata, 2021)

TATA deals with generating text based on information in charts, following the second task setup. This task has a long history, starting with work by Fasciano and Lapalme (1996), Mittal et al. (1998) and Demir et al. (2012), among others, who built modular chart captioning systems. But there are only a limited number of mainly English datasets to evaluate current models. These include Chart-to-Text (Obeid and Hoque, 2020; Kantharaj et al., 2022) consisting of charts from Statista paired with crowdsourced summaries and SciCap (Hsu et al., 2021), which contains figures and captions automatically extracted from scientific papers.

**Dealing with Noisy Data** While creating more challenging datasets is necessary to keep up with modeling advances, their creation process can introduce noise. Noise in simpler datasets can often be detected and filtered out through regular expressions (Reed et al., 2018), as done by Dušek et al. (2019) for the E2E dataset (Novikova et al., 2017). However, the larger output space in complex datasets requires more involved approaches and researchers have thus devised strategies to ensure that references are of sufficient quality. For example, ToTTo (Parikh et al., 2020) used an annotation scheme in which annotators were asked to remove non-attributed information from crawled text. SynthBio (Yuan et al., 2021) followed a similar strategy but started with text generated by a large language model (Thoppilan et al., 2022). The downside of involving crowdworkers in the language generation steps is that outputs can be unnaturally phrased compared to naturally occurring descriptions; studies on translationese in machine translation (Tirkkonen-Condit, 2002; Bizzoni et al., 2020) highlight potential negative effects on the final model and its evaluation (Graham et al., 2020). Our approach aims to mitigate these issues by transcribing naturally occurring descriptions.

---

parallel data published by the Demographic and Health Surveys (DHS) Program (https://dhsprogram.com/). Arabic, Hausa, Igbo, Swahili, and Yorùbá are spoken in the countries where the DHS conducts its surveys. These surveys are published alongside the colonial language spoken in these countries: English, French, and Portuguese. Russian was selected as an unseen test language.

[3] We use the terms *transcribe* and *transcription* as a shorthand for the process where the images of charts and diagrams (info-graphics) and their descriptions are manually converted by human annotators into spreadsheet tabular representations.

**Multilingual Generation** At present, there exist only few data-to-text datasets that cover languages beyond English (Gardent et al., 2017b; Kanerva et al., 2019; Dušek and Jurčíček, 2019). None of these datasets offers parallel data in two or more languages.[4] In contrast, TATA supports eight fully parallel languages focusing on African languages, and Russian as an additional zero-shot language. Each source info-graphic covers at least two of these languages; we provide data for the remaining languages using professional translations, leading to a fully parallel corpus while minimizing the drawbacks of only relying on translated texts.

Existing datasets available in African languages mainly focus on classic NLP applications such as machine translation (Nekoto et al., 2020) and summarization (Varab and Schluter, 2021). among others. TATA enables data-to-text generation as a new task for these languages and—due to its parallel nature—also supports the development of MT models that generalize to new domains and multi-modal settings (Adelani et al., 2022). In addition, due to the focus on surveys in African countries, the topics and entities in TATA are distinctly Africa-centric. This is in contrast to other African language datasets where Western-centric entities are over-represented (Faisal et al., 2022).

## 3 TATA

### 3.1 Desiderata and Overview

Our goal was to construct a challenging data-to-text dataset based on naturally occurring data i) that is not included in pretraining datasets and ii) which contains (preferably multiple) references that can be attributed to the input.

To fulfill these desiderata, TATA bridges the two communicative goals of data-to-text generation: i) It has medium-length inputs with a large output space, which allows studying content selection; and ii) it restricts generation to a single sentence at a time. The data is created through transcriptions of info-graphics and their descriptions found in PDF files, which ensures high quality references while avoiding training data overlap issues. Since each example can be described in multiple sentences, we select examples with the most sentences as test examples, assuming that they cover a larger share of the potential output space.

---

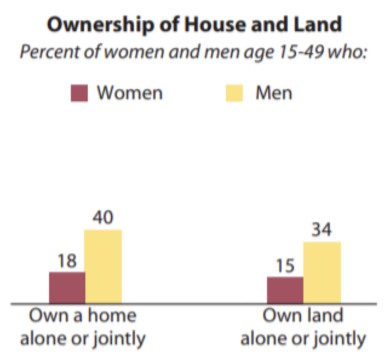

**Title** Ownership of House and Land
**Unit of Measure** Percent of women and men age 15-49 who:

|  | Women | Men |
| --- | --- | --- |
| Own a home alone or jointly | 18 | 40 |
| Own land alone or jointly | 15 | 34 |

**Linearized Form**: Ownership of House and Land | Percent of women and men age 15-49 who: | (Women, Own a home alone or jointly, 18) (Men, Own a home alone or jointly, 40) (Women, Own land alone or jointly, 15) (Men, Own land alone or jointly, 34)

**References**
1. Only 18% of women own a house, either alone or jointly, and only 15% own land.
2. In comparison, men are more than twice as likely to own a home alone or jointly (40%).
3. Men are also more than twice as likely to own land alone or jointly (34%).

Figure 2: An example of the process from infographic to linearized input. Each table value is encoded into a triple of (Column, Row, Value). The goal of the model is to generate text similar to the references below.

### 3.2 Data Collection

We extract tables from charts in 71 PDF reports published between 1990 and 2021 by the Demographic and Health Surveys Program[5], a USAID-funded program to collect and disseminate nationally representative data on fertility, family planning, maternal and child health, gender, and nutrition. The reports and included info-graphics are published in English and commonly a second language (Portuguese, French, Arabic, Yorùbá, Igbo, Hausa, and Swahili). 22 of the selected documents were only available in English, while 49 were bilingual, and the number of charts per document ranged from two to 97. A team of paid annotators transcribed the info-graphics into tables, in addition to extracting sentences referencing each chart in the

---

[4]An exception is the English–German RotoWire subset created for a shared task (Hayashi et al., 2019).

[5]https://dhsprogram.com/

| Language | # Transcribed / # Translated | Input Lengths / Output Lengths | F1 |
|---|---|---|---|
| Arabic | 157 / 711 | $952_{\pm 68}$ / $251_{\pm 9}$ | 0.23 |
| English | 903 / 0 | $369_{\pm 16}$ / $134_{\pm 4}$ | 0.16 |
| French | 88 / 778 | $470_{\pm 22}$ / $167_{\pm 6}$ | 0.18 |
| Hausa | 62 / 804 | $424_{\pm 20}$ / $160_{\pm 5}$ | 0.20 |
| Igbo | 32 / 834 | $455_{\pm 22}$ / $172_{\pm 5}$ | 0.24 |
| Portuguese | 23 / 833 | $453_{\pm 20}$ / $174_{\pm 6}$ | 0.18 |
| Swahili | 68 / 800 | $438_{\pm 20}$ / $154_{\pm 5}$ | 0.19 |
| Yorùbá | 25 / 841 | $662_{\pm 30}$ / $280_{\pm 10}$ | 0.31 |

Table 1: An overview of the TATA training data, including the number of transcribed/translated examples, input/output lengths based on the mT5 tokenizer (with 95% confidence interval), and Table F1 metric (§4.3). Since each language has the same tables, we can see that the tokenizer favors English since it has the shortest input lengths, while being the least compatible with Arabic and Yorùbá.

surrounding text.[6] Due to this process, the data in TATA is virtually unseen by pre-trained models, as confirmed in Section 4.

During the annotation, we also collected the following metadata: table ID (order of appearance in the report), page number (in the PDF), table title, unit of measure, and chart type (*vertical bar chart, horizontal bar chart, map chart, pie chart, table, line chart, other*). Each example additionally includes a screenshot of the original info-graphic. The extracted tables were then translated from English by professional translators to all eight languages, maintaining the table format. Each example includes a marker that indicates whether it was transcribed or translated. The final dataset comprises 8,479 tables across all languages.

To maximize the amount of usable data, we did not filter examples that lack associated sentences, and we also included examples in which the transcriptions do not include any values in the table (e.g., from bar charts without value labels). These examples were assigned to the training set and we explore in Section 4 how to use them during training. Figure 2 provides an example of the data.[7]

### 3.3 Data Splits

The same table across languages is always in the same split to prevent train-test leakage. In addition to filtering examples without transcribed table values, we ensure that every example of the development and test splits has at least 3 references. From the examples that fulfilled these criteria, we sampled 100 tables for both development and test for a total of 800 examples each. A manual review process excluded a few tables in each set, resulting in a training set of 6,962 tables, a development set of 752 tables, and a test set of 763 tables.

**Zero-Shot Russian** We further transcribed English/Russian bilingual documents (210 tables) following the same procedure described above.[8] Instead of translating, we treat Russian as a zero-shot language with a separate test set, selecting 100 tables with at least one reference.

### 3.4 Linearization

To apply neural text-to-text models to data-to-text tasks, the input data needs to be represented as a string. Nan et al. (2021) demonstrated in their DART dataset that a sequence of triplet representations *(column name, row name, value)* for each cell is effective in representing tabular data.[9] However, tables in TATA have between zero and two column and row headers while DART assumes homogeneously formatted tables with exactly one header. We thus additionally adopt the strategy taken by ToTTo (Parikh et al., 2020) and concatenate all relevant headers within the triplet entry, introducing special formats for examples without one of these headers. Similar to ToTTo, we append the table representation to the title and the unit of measure to arrive at the final representation, an example of which we show in Figure 2.

To identify the headers across the different table formats, we rely on a heuristic approach informed by our transcription instructions: we assume that the first $n$ rows in the left-most column are empty if the first $n$ rows are column headers (i.e., one in Figure 2). We apply the same process to identify row headers. If the top-left corner is not empty, we assume that the table has one row header but no column header; this frequently happens when the unit of measure already provides sufficient information. Our released data includes both the unmodified and the linearized representation, which we use

---

[6]Each annotator went through 2–3 rounds of training during which the authors provided feedback on 558 transcribed tables for correctness. Annotators relied on document parallelism to extract content for the languages they do not speak (and were familiar with the relevant orthographic system). We release all annotation instructions alongside the data.

[7]More examples shown in Appendix A.

[8]We selected Russian due to the number of available PDFs in the same format. We additionally considered Turkish, but found only four usable tables.

[9]While prior work (e.g. Wiseman et al., 2017) used a similar representation, DART was the first larger study on how to represent tables.

throughout our experiments.

## 3.5 Dataset Analysis

Table 1 provides an analysis of the training split of our data. The data in each language is comprised of transcriptions of 435 vertical bar charts, 173 map charts, 137 horizontal bar charts, 97 line charts, 48 pie charts, 5 tables, and 9 charts marked as "other". On average, a table has $11\pm16$ cells (not counting column and row headers), with a minimum of one and a maximum of 378 cells. Due to varying tokenizer support across languages, the linearization lengths of inputs and even the output lengths have a very high variance. To ensure that targets in the dataset are unseen, we measured the fraction of TATA reference sentences that overlap with mC4 (Xue et al., 2021) at the level of 15-grams as 1.5/1.7% (dev/test).[10] This validates that our data collection approach produced novel evaluation data that is unlikely to have been memorized by large language models (Carlini et al., 2022).

# 4 Experiments

## 4.1 Setup

We study models trained on TATA in three settings: monolingual, cross-lingual and multilingual. We train *monolingual* models on the subset of the data for each language (8 models) and evaluate each model on the test set of the same language. The *cross-lingual* setup uses these models and evaluates them also on all other languages. The *multilingual* setup trains a single model on the full training data. If a training example has multiple references, we treat each reference as a separate training example.[11] In the multilingual setup, we compare different strategies for dealing with incomplete data:[12]

**Missing References**   To handle examples with missing references, i.e., examples for which no verbalizations were available to transcribe, we compare two strategies. First, we simply do not train on these examples (SKIP NO REFERENCES). Second, we use a tagging approach suggested by Filippova (2020) where we append a "0" to the input for examples without a reference and learn to predict an

empty string. For examples with references, we append "1". We then append "1" to all dev and test inputs (TAGGED).

**Missing Table Values**   Since inputs from tables with missing values will necessarily have non-attributable outputs, we investigate two mitigation strategies. To remain compatible with the results from the above experiment, we base both cases on the TAGGED setup. First, we filter all examples where tables have no values (SKIP NO VALUES). We also take a stricter approach and filter references whose content has no overlap with the content of the table based on our *Table F1* metric (see Section 4.3), denoted as SKIP NO OVERLAP.

## 4.2 Models

We evaluate the following multilingual models.

**Multilingual T5** (mT5; Xue et al., 2021) is a multilingual encoder-decoder text-to-text model trained on Common Crawl data in 101 languages. To assess the impact of model scale, we evaluate it in both its small (mT5$_{\text{small}}$; 300M parameters) and XXL (mT5$_{\text{XXL}}$; 13B parameters) configurations.

**SSA-mT5** As African languages are underrepresented in mT5's pre-training data, we additionally evaluate a model that was pre-trained on more data in Sub-Saharan African (SSA) languages. The model was trained using the same hyper-parameters as mT5, using mC4 (Xue et al., 2021) and additional automatically mined data in around 250 SSA languages (Caswell et al., 2020). We only use the small configuration (mT5$_{\text{SSA}}$; 300M parameters) for this model.

## 4.3 Evaluation

**Human Evaluation** Automatic metrics are untested for many of our languages and the setting of TATA, and outputs may still be correct without matching references (Gehrmann et al., 2022). Our main evaluation is thus through having expert human annotators judge model outputs in a direct-assessment setup where they have access to the input table. We evaluate one (randomly sampled) reference and three model outputs for every development and test example. Model outputs are from multilingually trained models. We report results on the test set, while we use the annotations of the development set to create an automatic metric.[13] All evaluation annotators are fluent in the respective

---

[10]For comparison, the estimate for relevant languages in the widely used Universal Dependencies 2.10 treebanks (De Marneffe et al., 2021) with mC4 is 45% (averaged over Arabic, English, French, Portuguese, Russian and Yorùbá).

[11]For hyperparameters, see Appendix B.

[12]These strategies can also be applied to the monolingual settings, but we omit such experiments for brevity and focus on the highest-performing (multilingual) setting.

---

[13]Both sets will be publicly released.

languages and were instructed in English.[14]

To maximize annotation coverage, outputs were only evaluated one-way and we are thus unable to provide inter-annotator agreement numbers. However, all instructions were refined through multiple piloting and resolution rounds together with the annotators to ensure high quality. Moreover, in a comparison with internal gold ratings created by the dataset authors, the annotators achieve an F1-Score of 0.9 and 0.91 for references and model outputs, thus closely tracking the "correct" ratings.

Each sentence is annotated for a series of up to four questions that implement a task-specific variant of Attribution to Identifiable Sources (AIS; Rashkin et al., 2021) and enquire about understand-ability, attributability, required reasoning, and cell coverage (see Appendix D for details).

**Automatic Evaluation** We investigate multiple automatic metrics and use the human annotations to assess whether they are trustworthy indicators of model quality to make a final recommendation which metric to use. (1) **Reference-Based**: We assess {P,R,F}-score variants of ROUGE-{1,2,L} (Lin, 2004) as n-gram based metric, CHRF (Popović, 2015) with a maximum n-gram order of 6 and $\beta = 2$ as character-based metric, and BLEURT-20 (Sellam et al., 2020; Pu et al., 2021) as learned metric. We compute the score between the candidate and each reference and take the maximum score for each table. (2) **Reference-less (Quality Estimation)**: We define TABLE F1, which compares a candidate and the input table. We create a set of tokens contained in the union of all cells in the table, including headers ($R$), and the set of tokens in a candidate output ($C$). From there, we can calculate the token-level precision ($\overline{R \cap C}/\overline{C}$), recall ($\overline{R \cap C}/\overline{R}$), and F1-score. This can be seen as a very simple, but language-agnostic variant of the English information matching system by Wiseman et al. (2017). (3) **Source and Reference-Based**: We use PARENT (Dhingra et al., 2019), which considers references and an input table. As it assumes only a single level of hierarchy for column and row headers, we concatenate all available headers, and collect PARENT-R/P/F.

For all metrics that require tokenization (ROUGE, TABLE F1, PARENT), we tokenize references, model outputs, and table contents using the mT5 tokenizer and vocabulary.

| Setting | nU – U – U+A | Reasoning | # Cells |
|---|---|---|---|
| Reference | | 0.40 / 0.75 | $8.0_{6.7}$ |
| mT5$_{small}$ | | 0.03 / 0.78 | $6.9_{5.9}$ |
| mT5$_{SSA}$ | | 0.03 / 0.77 | $6.8_{5.1}$ |
| mT5$_{XXL}$ | | 0.34 / 0.77 | $7.9_{6.0}$ |

Table 2: Results from the human evaluation aggregated over all languages. Left is the distribution of not understandable (**nU**; red), understandable (**U**; grey), and understandable and attributable (**U+A**; green). Right, we show the fraction of examples marked as demonstrating reasoning compared to all examples and as fraction of U+A examples. The rightmost column shows how many cells were reasoned over (with standard deviation). The references and the XXL model both achieve high U+A rates of 0.53 and 0.44 respectively. Note that only one reference was evaluated per example. Surprisingly, the reasoning extent is very similar across all models if we focus on only good outputs.

**STATA** As additional metrics, we fine-tune mT5$_{XXL}$ on the human assessments of model outputs and references in the development set. To do so, we construct the metric training data by treating all understandable + attributable examples as positives and all others as negative examples. We adapt mT5 into a regression-metric by applying a RMSE loss between the logits of a special classification token and the label which is either 0 or 1. During inference, we force-decode the classification token and extract its probability.[15]

We denote this metric *Statistical Assessment of Table-to-Text in African languages*, or STATA. We train three STATA variants that follow the setups of the previously introduced automatic metrics: as quality estimation model that predicts a score based on the table input and the model output without references (*QE*), with references (*QE-Ref*), and as a traditional reference-based metric (*Ref*).

## 5 Results

### 5.1 Human Evaluation

The human evaluation results in Table 2 show that only 44% of annotated samples from mT5$_{XXL}$ were rated as both understandable and attributable to the input. This means that TATA still poses large challenges to models, especially small models, since even the best model fails 56% of the time and the smaller models most of the time are not

---

[14]Instructions are available in Appendix J.

[15]More details are in Appendix C.

| | Correlation with U+A |
|---|---|
| BLEURT-20 | 0.12 |
| ROUGE-1 P/R/F | 0.07 / 0.09 / 0.11 |
| ROUGE-2 P/R/F | 0.12 / 0.11 / 0.13 |
| ROUGE-L P/R/F | 0.08 / 0.11 / 0.13 |
| TABLE P/R/F | 0.02 / 0.06 / 0.05 |
| CHRF | 0.16 |
| STATA QE | **0.66** |
| STATA QE+REF | 0.61 |
| STATA REF | 0.53 |

Table 3: Pearson Correlations between metrics and the U+A human ratings.

understandable. This finding is consistent across all languages (see Appendix E).

Our annotated references perform better across these quality categories, mostly failing the attribution test when the transcribed sentences include unclear referential expressions or additional information not found in the infographic (see references 2 and 3 in Figure 2). However, since only one of the 3+ references was annotated, the probability of an example having at least one high-quality reference is high. Interestingly, of the examples that were rated as attributable, over 75% of sentences from all models require reasoning over multiple cells, and the number of cells a sentence describes closely follows the number from the references.

We further performed a qualitative analysis of 50 English samples to identify whether a sentence requires looking at the title or unit of measure of a table. While 1/3 of references follow or make use of the phrasing in the title, and 43% of the unit of measure, for mT5$_{XXL}$, the numbers are 54% and 25%—staying closer to the title while relying less on the unit of measure.

## 5.2 Existing Metrics are Insufficient

Following our initial hypothesis that existing metrics are untested and may not be suitable for TATA, we conduct a correlation analysis between the human evaluation ratings and metric scores. Table 3 shows the result of comparing to the main desired outcome: whether a sentence is understandable and attributable. Existing metrics perform very poorly at this task, with a maximum correlation of 0.16 for chrF. This confirms that comparing to a set of references fails to detect non-understandable and non-attributable outputs, but even the TABLE F1 metric, which is reference-agnostic falls short.[16]

This finding is intuitive as these metrics were not designed to evaluate the correctness of reasoning. Nevertheless, they are used to assess outputs in recent data-to-text approaches (e.g., Mehta et al., 2022; Yin and Wan, 2022; Anders et al., 2022), although most point out the limitations of such automatic assessments.

Performing the correlation analysis using only understandability as target, which is a much easier task for a metric, leads to only slightly improved results, with BLEURT-20 having a correlation of 0.22, while all remaining metrics are at or below 0.13. The results for the reasoning and cell count questions are similarly poor, with maximum correlations of 0.09 and 0.18 respectively.

The dataset-specific metric STATA fares much better. Similarly, comparing outputs to references does not contribute to an improved correlation and Quality Estimation is the best setup, achieving a correlation of 0.66, which in this case is equivalent to an AUC of 0.91. This correlation is on par with the agreement between the raters and our internal gold-annotations. Our experiments further showed that it was necessary to start with a *large* pre-trained model: Training in the QE setup starting from an mT5$_{base}$ only achieved a correlation of 0.21, only slightly outperforming existing metrics.

As a result of these findings, we mainly report results with STATA in the next section (see Appendix G for results with other metrics). We report chrF as the best performing existing metric, but plead caution in interpreting its numbers.

## 5.3 Automatic Evaluation

We show the automatic evaluation results in Table 4. Similar to the findings from the human evaluation, the model pre-trained on additional data in African languages slightly outperforms the standard mT5$_{small}$ model, but neither get close to the performance of mT5$_{XXL}$, demonstrating the impact of scale even for under-represented languages. Similarly, all multilingually trained models outperform monolingual training. The multilingual settings perform similar to each other, with SKIP NO REFERENCES and SKIP NO OVERLAP leading to the highest scores. While chrF correctly ranks the XXL models above the others, but only with a very minor margin, and it fails to distinguish between monolingual and multilingual training setups.

---

[16]Since PARENT reports aggregate scores over the entire test corpus, we cannot compute the segment-level correlation, but we found similarly poor performance when assessing it on the system-level.

| Setup | Model | CHRF | STATA QE | STATA REF | STATA QE+REF |
|---|---|---|---|---|---|
| MONOLINGUAL | SSA | 0.37 | 0.04 | 0.50 | 0.25 |
| | Small | 0.36 | 0.03 | 0.49 | 0.25 |
| | XXL | **0.39** | 0.28 | 0.62 | 0.44 |
| SKIP NO REFERENCES | SSA | 0.35 | 0.05 | 0.51 | 0.26 |
| | Small | 0.33 | 0.01 | 0.47 | 0.23 |
| | XXL | **0.40** | **0.61** | **0.76** | **0.74** |
| TAGGED | SSA | 0.35 | 0.09 | 0.54 | 0.29 |
| | Small | 0.34 | 0.07 | 0.52 | 0.27 |
| | XXL | **0.41** | 0.57 | **0.76** | 0.69 |
| + SKIP NO VALUES | SSA | 0.37 | 0.11 | 0.57 | 0.32 |
| | Small | 0.34 | 0.10 | 0.54 | 0.30 |
| | XXL | **0.40** | 0.55 | **0.76** | 0.71 |
| + SKIP NO OVERLAP | SSA | 0.32 | 0.01 | 0.47 | 0.22 |
| | Small | 0.28 | 0.00 | 0.42 | 0.19 |
| | XXL | **0.39** | **0.59** | **0.77** | **0.75** |

Table 4: Evaluation Results. MONOLINGUAL represents the average score of the in-language performances of separately trained monolingual models. All others are multilingually trained models and we average over their per-language scores. SKIP NO REFERENCES omits training examples without references while TAGGED uses a binary indicator in the input whether an output is empty. The final two variants build on TAGGED to additionally filter out training examples where table values are missing or where a reference has no overlap with any value in the table. For each column we bold-face the highest results (including those that are not significantly different from them). According to STATA, the largest gains come from scaling to larger models and both SKIP NO REFERENCES and SKIP NO OVERLAP outperform the other modalities.

**Cross-lingual** We show and discuss the monolingual and cross-lingual performance of models using STATA QE in Appendix F. Our main observations are: **i)** English is far from the best source language; **ii)** Swahili is the best source language, likely due to its strong linguistic connections to other languages in our data; and **iii)** models perform on par for tonal languages like Hausa and Yorùbá, which may be due to the government report data using romanized orthography for Hausa, which omits tone and vowel length information (Schuh and Yalwa, 1993) and may thus facilitate cross-lingual transfer. In contrast, if we had relied on the cross-lingual results using standard metrics in Appendix G and H (for zero-shot Russian), we would have been led to very different conclusions. For instance, the standard metrics fail to identify the weak results of smaller models and mistakenly present Hausa and Yorùbá as the strongest languages. This discrepancy highlights the need for good metrics.

**Failure Cases** We observe a qualitative difference between the smaller and the large model outputs. The smaller models very commonly fail at parsing the table and generate nonsensical output like "*However, the majority of women age 30-49 are twice as likely to be fed the first birth.*" in the context of ages at first birth. In addition to complete failure, the model generates "majority" and "twice

as likely" in the same context, showing that it has not learned the correct associations required for reasoning. Moreover, many of the non-understandable examples suffer from repetitions and grammatical mistakes as in "*However, the majority of women age 30-49 have a typical of births and the lowest percentage of women who have a twice as high as a typical of births.*"

In contrast, the large models rarely fail at such fundamental level and instead sometimes generate dataset artifacts that include generalizations like "*The results show that the earlier start of family formation is very similar to the typical pattern.*" Another issue that arises in large models are outputs in which the reasoning is correct, but stated in a very clumsy way, as in "*Women are least likely to own a home alone or jointly with a partner, as compared with 34% of men*". More examples can be found in our released human annotations.

## 6 Conclusion

In this paper, we introduce TATA, a table-to-text dataset covering nine different languages with a focus on African languages and languages spoken in Africa. TATA is the first multilingual table-to-text dataset among many existing English-only datasets and it is also fully parallel, enabling research into cross-lingual transfer. We experiment with differ-

ent monolingual and filtered and augmented multilingual training strategies for various models. Our extensive automatic and human evaluation identifies multiple avenues for future improvements in terms of understandability, attribution, and faithfulness of neural generation models and their metrics. We develop the metric STATA based on additional data collected on the development set and demonstrate that it has a much better agreement with human ratings than existing metrics, which we consider unreliable and which we show lead to misleading results when analyzing transfer between languages.

## Limitations

Firstly, while we transcribed all available tables in their language, the majority of the tables were published in English as the first language. We use professional translators to translate the data, which makes it plausible that some translationese exists in the data. Moreover, it was unavoidable to collect reference sentences that are only partially entailed by the source tables, as shown in Table 2. Since our experiments show that additional filtering does not lead to improved performance, we are releasing the dataset as-is and encourage other researchers to investigate better filtering strategies. Moreover, we treat STATA QE as the main metric for the dataset, which is agnostic to references and should thus be more robust to the noise.

We finally note that the domain of health reports includes potentially sensitive topics relating to reproduction, violence, sickness, and death. Perceived negative values could be used to amplify stereotypes about people from the respective regions or countries. We thus highlight that the intended academic use of this dataset is to develop and evaluate models that neutrally report the content of these tables but not use the outputs to make value judgments.

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

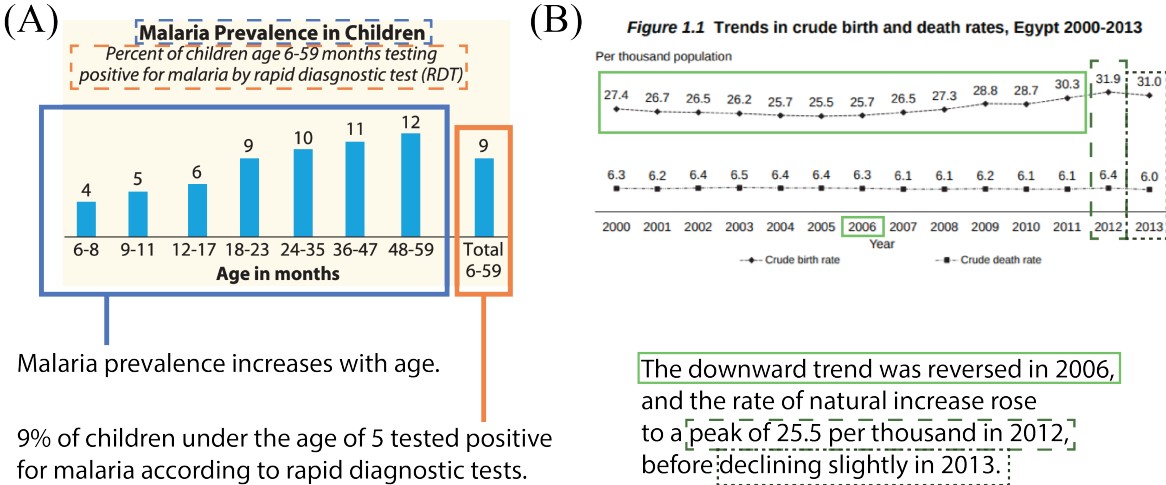

Figure 3: Two example info-graphics and their associated descriptions. Colored rectangles indicate where information from the text can be found in the figure. (A) The first sentence compares all numbers except the aggregate and infers that the numbers are increasing. The second sentence does not require any reasoning, but requires the inference that 6–59 months can be stated as "under the age of 5". (B) This sentence requires identifying the overall trend and calculating the peak population increase as the difference between birth and death rate ($31.9 - 6.5 = 25.5$). In addition to the values, sentences across both examples require accessing the title, unit of measure, or axis labels.

## A  Additional Examples

Figure 3 provides two additional examples of reasoning challenges in TATA.

## B  Details on Model Training

We train all models With a constant learning rate of 0.001 and dropout rate of 0.1 for all tasks, following the suggestions by Xue et al. (2021). During training, we monitor the validation loss every 25 steps for a maximum of 5,000 steps and pick the checkpoint with minimum loss. While the XXL model commonly converges within 100-200 steps, the smaller models often require 2,000+ steps to converge.

## C  Details on STATA

We use mT5-XXL (Xue et al., 2021) as base model which we finetune for 2,500 steps with a batch size of 32 using a constant learning rate of 1e-4. Inputs are truncated to a maximum length of 2048. We add the following inputs depending on the metric type: If the metric uses the input, we use the linearized representation of the example following a tag [source]. If the metric uses the references, we sample three of them for consistency, and add them after [reference] tags. The output always follows a [candidate] tag.

## D  Human Evaluation Details

Each sentence is annotated for a series of up to four questions. The first two questions ask whether annotators agree with binary statements, implementing a variant of "Attribution to Identifiable Sources" (AIS; Rashkin et al., 2021). The first asks whether a sentence is overall understandable by an annotator, allowing minor grammatical mistakes.[17] If the answer is no, the annotation of the example is complete. Otherwise, the task proceeds to the next question, which asks whether *all* of the information in the sentence is attributable to the table or its meta information, i.e., whether every part of the model output is grounded in the input. A single mistake (e.g., number or label) means that the sentence is not attributable, the only exception being minor rounding deviations (e.g., *"two thirds"* instead of *"65%"*). If the answer to the second question is no, the task terminates; otherwise, we ask two final questions.

The third question asks whether the generated text requires reasoning or comparison of two or more cells (*"X has the highest Y"*, or *"X has more Y than Z"*), and the last question asks annotators to count the number of cells one has to look at to generate the information in a given sentence.

---

[17]The exact definition we use is *"A non-understandable description is not comprehensible due to significantly malformed phrasing."*

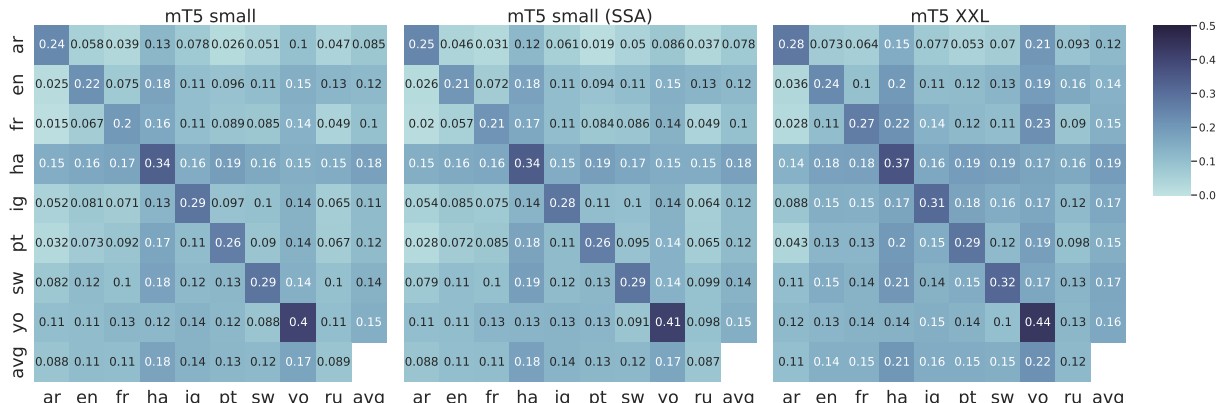

Figure 4: Cross-lingual zero-shot transfer performance of different monolingual models across all language pairs using **standard metrics**. Each value represents an average over the traditional metrics for a model trained on one language (rows) and evaluated on another one (columns). The final row/column represent an average. As expected, the highest values are along the within-language diagonal, but we also observe some curious behavior for Hausa and Yorùbá and in general large disagreements with the numbers presented in Figure 5.

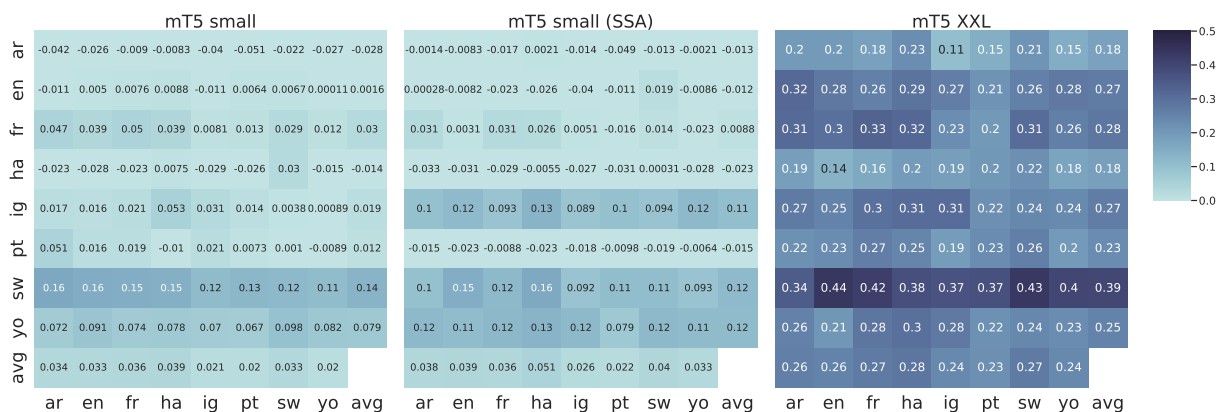

Figure 5: Cross-lingual zero-shot transfer performance of different monolingual models across all language pairs. Each value represents the STATA QE metric for a model trained on one language (rows) and evaluated on another one (columns). The final row/column represent an average.

## E  Detailed Human Evaluation Results

Table 8 presents the detailed human evaluation results by language. We can observe that there is some variation between languages (e.g., All examples in Yorùbá were judged as using reasoning), which we attribute to different understanding of the instructions for annotators in different languages. As a result, we suggest not comparing results across languages but instead focusing on the between-model comparison for a given language.

Focusing on this, the results are surprisingly consistent. The two smaller models have extremely low scores for the first two questions while the XXL-sized model follows the references with a small margin between the two. There is significant room for improvements on the task since success would mean being close to 1.0 for understandable

and attributable, which no model achieves.

While the references are not perfect either, STATA does not use them in the QE setting, and TATA is thus a fitting testbed for learning from somewhat noisy labels. We further note that the results for reference represent only one reference out of the 3+ available and there is thus a high probablity of the *set of references* to paint a more accurate picture of the output space for a table.

## F  Cross-lingual Results with StATA

We present the monolingual and cross-lingual performance of models trained on every language individually in Figure 6. The figure shows the STATA QE score for each training language (rows) evaluated on each target language (columns), along with averages.

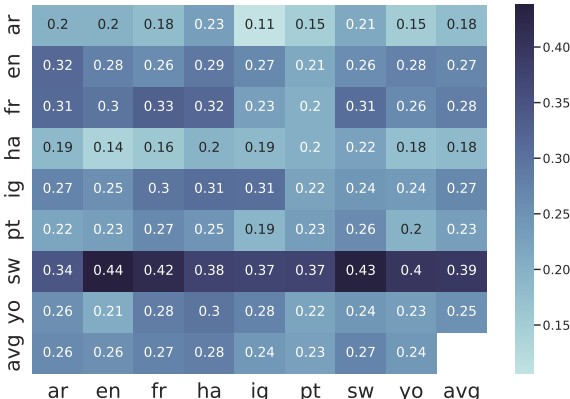

Figure 6: Cross-lingual zero-shot transfer performance of different monolingual models across all language pairs. Each value represents the STATA QE metric for an XXL model trained on one language (rows) and evaluated on another one (columns). The final row/column represent an average.

We make three key observations. First, it is evident that English is far from the best source language despite its prevalence in the pretraining corpus. This validates our design choice to avoid English-centric data during the collection of TATA, and points to future work to collect data specific to non-Western locales. Second, Swahili achieves the highest performance across almost all transfer scenarios. Swahili has a substantial shared vocabulary with English, Arabic, and Portuguese, and it is distantly related to Igbo and Yorùbá. These strong linguistic connections likely explain the observation. However, Swahili even transfers well to Hausa, which is both geographically and genetically distant from Swahili (Chadic vs. Volta-Congo/Niger for the others).

Finally, in cross-lingual transfer between the other African languages, we observe some weak effects of their geographic distribution or genetic relationship, in line with previous findings that the geographic location of speakers and linguistic similarity between two languages are indicative of positive transfer (Ahuja et al., 2022). We would expect positive transfer effects between Hausa, Igbo, and Yorùbá, due to their proximal geographic distribution in Western Africa, and even stronger effects between Igbo and Yorùbá, which are closer related to each other than Hausa. But, we do not see strong evidence for this relationship.

Additionally, while tonal languages like Hausa and Yorùbá are not well represented in pretraining data and tokenizers (Alabi et al., 2020; Adebara and Abdul-Mageed, 2022), the models perform on

| Setup | Model | CHRF |
|---|---|---|
| SKIP NO REFERENCES | SSA | 0.17 |
| | Small | 0.18 |
| | XXL | **0.34** |
| TAGGED | SSA | 0.18 |
| | Small | 0.18 |
| | XXL | 0.23 |
| + SKIP NO VALUES | SSA | 0.17 |
| | Small | 0.15 |
| | XXL | 0.27 |
| + SKIP NO OVERLAP | SSA | 0.17 |
| | Small | 0.18 |
| | XXL | 0.24 |

Table 5: Noisy chrF test results on zero-shot transfer to Russian for multilingually trained models.

par as for other languages with a slight edge for Hausa. This edge could be explained by the fact that TATA is based on government reports that use the romanized orthography for Hausa, which omits tone and vowel length information (Schuh and Yalwa, 1993), and which may thus facilitate cross-lingual transfer.

The cross-lingual peformance numbers also further emphasize the need for good metrics. We present an extended version of the figure using an average of all existing metrics and with STATA in the next section in Figures 4 and 5. If we had relied on the existing metrics, we would have been led to very different conclusions.

## G  Cross-Lingual Results with Different Metrics

Figures 4 and 5 show the detailed cross-lingual results when relying on standard metrics (4) compared to STATA (5). For the standard metrics, we present an average of our baseline metrics for each (model, language) pair. The standard metrics fail to identify the weak results of the smaller models and mistakenly present Hausa and Yorùbá as the strongest languages.

## H  Zero-shot Evaluation to Russian

Since there is no training data for STATA in Russian, we leave the in-depth zero-shot evaluation in a distant language for future work and focus on noisy chrF numbers in Table 5. While differences between setups were small using chrF in Table 4, for zero-shot transfer to a new language SKIP NO REFERENCES seems to perform best with a significant margin.

# I  Transcription Instructions

The following are the instructions that were presented to annotators during transcription. Note that there are multiple references to spreadsheets which record links to the external documents and to locations where annotators can enter the transcriptions. Every step of instructions was additionally accompanied by screenshots we are unable to share.

## I.1  Overview

You will be accessing two PDF documents with similar content and structure. One will be in English and the other one is a version of the document in a different language. You don't need to know the other language for this task, English is enough. Starting in English, you will work from the start of the document and find the charts, convert them into a table using Google Sheets, extract any text that refers to the chart, and then repeat the process in the other language. You will also take a screenshot of both charts and save them to Google Drive.

We're interested in the outcome as well as in the process: please let us know if anything in this document is unclear and what challenges you encounter following the steps below. If you would like to leave specific comments on a particular table (eg. doubts, questions, something you were unsure of), kindly add them in column S "Comments" in the Table index.

## I.2  Cheat Sheet

- Claim your document assignment from this list

- Open the PDF files and tables.

- Find the first chart in the English document and transpose into a table in the spreadsheet by filling the template in Tab 1.

- Take a screenshot of the chart, upload it to the folder and rename it to match the name of the table.

- Add the text from the Table.

- Repeat steps 2-5 for the same chart in the second language.

- Find the next chart in English and repeat steps 2-6.

  Always remember to:

- Check the spelling, especially when transcribing a language you do not speak.

- Check that the number of the table ID corresponds with the name of the tab.

## I.3  Creating your first Table in English

1. The documents you will extract the data from are in PDF format. You will be working on one document at a time. Claim the document you will be working from this the Document Index by adding your user name to the "Claimed by" cell of one of the rows

2. Now open the PDF documents listed in columns C and E of the same row. There can be two different PDFs, one per language, or just one, where the content in the second language comes after the content in the first language. If this is the case, you will see the same text in columns C and E.

3. Now, open the spreadsheets linked from columns D and F. This is where you'll create the tables. The two spreadsheets have the same alphanumerical name with a different two letter code at the end to differentiate between English ("en") and the second language. The spreadsheets have been pre-populated with templated tabs for you to transform your chats into. The tabs are named with numbers starting at 1. You'll transform the first chart into tab 1, the second chart into tab 2, the third chart into tab 3, and so on. To access tabs with higher numbers, click the right arrow at the bottom right of Spreadsheets (and then the left one to return to the original view)

4. Templates include the following fields for you to fill in:

   - *Table ID*: refers to the order in which the chart appears in the document: the first chart will be 1, the second chart will be 2 and so on. This should match with the tab number in the spreadsheet into which you extract the chart's information.
   - *Page Number*: this is the page number where the chart appears. Do not trust the page numbers written inside the document. Use the page number shown in the PDF viewer (see example)
   - *Title*: This is the title of the chart as in the document

- *Unit of measure*: include the chart / table's unit of measure here
- *Screenshot link*: You will take a screen capture of the chart and enter the link here (instructions below).
- *This is how you capture your screen*: [Omitted for brevity]

5. Now it's time to look for charts. Start skimming the PDFs from the beginning of your English text looking for charts and tables. We are interested in all types of charts: bar charts, pie charts, map charts, etc. (see chart examples below). The only exception to this rule is HIV related information: please skip any HIV-related charts, these will not be transcribed. Non-exclusive list of HIV-related examples:

   - HIV Prevalence by Marital Status
   - Trends in Recent HIV Testing
   - HIV Prevalence by Province
   - Trends in HIV Prevalence

   The next steps to transcribe the chart require some more thinking: please interpret at a basic level what the English chart is about and what information it conveys. It won't work well to copy-paste it blindly.
   You can then start populating your table. Enter the Table ID, the Page Number, the chart Title, the Unit of measure and the screenshot link as explained above.
   Please enter the table below the line START OF TABLE «Enter table contents below», leaving this text unchanged. Note you may need to add rows to fit in all the information from the chart.
   Then identify the axes in the chart and start populating the table with row and column headers: copy the text exactly as it appears in the PDF. As best you can, try to include the item with the largest number of items in rows, so that the table is taller (vertical) rather than wider (horizontal). You do not have to duplicate the unit of measure in the table if there are other headers already (see example to the right, and 4-5 below). If the unit of measure is the same as the column header (see examples 1-3 below), include it as column header.
   One example is shown here (image below, table to the right) and for other charts on the final page of this document.

6. Once all the chart text has been added to row and column headers, fill in the table with values from the chart.

7. Now look in the text surrounding the chart to find text that refers to the chart. It can be anything explicitly mentioning the data in the table. In the document, you will normally find it before or after the chart. We are interested in capturing only those sentences that describe or compare the chart's data, so watch out for irrelevant sentences appearing mid-paragraph, as illustrated here. [This is a common pitfall] When you find the text you'll add it to the table below the line "START OF TEXT «Enter text below. Move this row further down if the table needs more space»". You'll add the text sentence by sentence, one sentence per cell in the first column only, as shown in the following example. Make sure to include full sentence casing and punctuation, as in the original text.

8. Once you have added all the text, add your new table to the Table Index next to your screenshot (step 4): [details omitted for brevity]

CONGRATULATIONS! You have completed all the steps for the first English chart. Now it's time to move to the second language.

## I.4 CREATING YOUR FIRST TABLE IN THE SECOND LANGUAGE

1. In the PDF for the second language (or the section of the bilingual document you're working from that corresponds to the second language), find the same chart you just transformed into a table. The documents are aligned in both languages, so they should be in the same page in both documents.
   When you find the chart, repeat steps 4 to 10 using the second language table in the document index and replacing where it says English (EN) by the two letter code of your second table (e.g. "SW" for Kiswahili).
   Tips to working in a second language:

   - Documents are aligned by language. This means that the formatting of the text in one language mirrors the second language. Therefore, when looking for the same chart, go to the same page in

the second document if you have two documents. Or find the start of the second language if you're working with one bilingual document, and use images and charts to guide you until you find what you're looking for.

- Check the number of sentences in the paragraph by identifying the full stops. This will help you find the start of the sentence you'd like to copy-paste.
- Geographical names may be similar or related: you can use them to confirm you're copying the sentence you wanted to copy.
- Identify the numbers in the sentence: you can use them to confirm you're copying the sentence you wanted to copy.
- **Arabic**: Arabic is a language with a right-to-left writing system (in English, the words are written left-to-right). As you go through Arabic text, remember that sentences start on the right and continue to the left. Also note the cursor of your mouse may behave strangely when selecting the text for copy pasting. You may need to start selecting on the right-hand side, and drag towards the left.

WELL DONE! You have two aligned tables, one in English and one in a different language. Now continue adding more tables.

## I.5 ADDING MORE THAN ONE TABLE TO YOUR SPREADSHEETS

After completing your first table in English and your second language, it's time to continue looking for the second chart back in the English document. When you find the next chart in the document, fill out the next tab (e.g. 2) by selecting it at the bottom of your screen.

Now repeat steps 4-10 for English, adding a new line to the table index, and again for the second language until you reach the end of the PDF. At this point, your table should have as many tabs as the number of tables in column F of the Document index.

If anything in these guidelines is unclear, please let us know. Thanks!

## I.6 CHART AND TABLE EXAMPLES

[5 examples here omitted for brevity]

## I.7 COMMON PITFALLS

[Note: Every pitfall is accompanied by a screenshot or an example.]

**Ensure that all copied text is relevant to the table.** **Irrelevant Text**: Stunting also varies by governorate. Stunting is below 25% in Aden, Abyan, and Al Mhrah, and is highest in Reimah, at 63%. **Relevant Text**: According to the survey, 47% of children under five are stunted, or too short for their age.

→ Please only copy sentences that are directly supported by the chart which means that numbers in the table or its column/row names or title are directly referenced or compared. Should a sentence partially be supported by a table, please still copy it in full.

**In Arabic, ensure that entire sentences are copied, and that only the relevant sentences are copied.** Periods are hard to spot, but if you are unsure about where the sentence ends you can double check by sending a copied sentence through Google Translate:

**Make sure to also transcribe map charts** In map charts, every region should be a separate row.

**Do not leave empty rows and columns for spacing** Here we have two empty rows under the table before the start of the text, highlighted in yellow. These rows should be deleted (right-click on your mouse to quickly access this option). Empty spaces make it difficult to read the tables automatically.

**Ensure number formatting matches the original values** When entering percentages (e.g. 10%) or ranges (e.g. 10-20) into Google Sheets, they may get reformatted automatically and no longer match the original values. **Watch out for this**. Fix by adjusting the number formatting inside the Sheet until your table matches the original document's values. If copy-pasting from the document, try to paste without formatting (Ctrl+Shift+v).

**Representation of additional graph elements** Some charts are designed in such a way that they contain additional elements. When representing them, keep the below in mind:

**Secondary scale annotations** Add a column to the left of the main categories and fill out the values as they correspond in the graph.

**Supercategories** Identify supercategories in a separate row with the value cell left empty. Add all

subcategories of a given supercategory below.

**Representation of Footnotes** When a table or its text includes footnotes, you do not need to add the footnotes to the spreadsheet. We additionally ask you to try to remove the footnote symbol from the text itself.

**Use the PDF page numbers, not the ones written in the document** Please use the page numbers indicated by the PDF reader and not those appearing inside the document.

## J Human Evaluation Instructions

The following are the instructions that were presented to annotators in our human evaluation.

### J.1 Overview

In this task you will evaluate the quality of one-sentence descriptions of a table. You will evaluate multiple sentences for the same table that are all independent of each other. Each descriptive sentence should make sense and be grounded in information provided in the source table.

For each sentence, we ask you to rate along four dimensions:

1. The text is understandable

2. All of the provided information is fully attributable to the table, its title, and its unit of measure.

3. How many cells does the text cover?

4. Generating the text requires reasoning or comparison of multiple cells.

The first three dimensions are binary statements where we ask you to answer with a "No" (false) or "Yes" (correct) and the last one asks you to count the number of cells that a description covers. The sections below describe each of the dimensions in detail.

### J.2 Additional notes

The descriptions may appear very fluent and well-formed, but can contain inaccuracies that are not easy to discern at first glance. Pay close attention to the table. If you are unsure about any particular answer, please enter "-1" in the relevant cell.

**(Q1) The text is understandable.** In this step you will evaluate whether you can understand the sentences on their own. You may consult the table

for this stage in case that context is required, but you should ignore all other sentences when making your judgment. Carefully read the sentences one-by-one and decide whether you agree with the following statement: "The text is understandable".

**Definition**: A non-understandable description is not comprehensible due to significantly malformed phrasing.

The purpose of Q1 is to filter out descriptions that you cannot rate along the other dimensions because you cannot understand their meaning. If the description is unclear, select "No". In other words, if you cannot understand what a sentence is trying to say to the extent that you will not be able to rate it along the other dimensions, mark it as "No."

In the case a sentence has conjunctions that don't make sense for alone-standing sentences (e.g., starting with "However", you can still mark it as understandable if the rest of the sentence makes sense.

Please do not mark anything as "No" that is factually incorrect, making value judgments. The question is only meant to filter out sentences like "The proportion of women declined by 30%" or "Women proportion by 30%" where it is completely unclear what the text refers to or that are so ungrammatical that it becomes nonsensical.

*If you select "No", you do not have to answer the remaining questions.*

**(Q2) All of the provided information is fully attributable to the table, its title, and its unit of measure.** If a sentence is understandable, we next ask you to read the associated table and its metadata above it. Then, mark whether you agree with the above statement for each individual sentence. You should write "Yes" only if all of the information provided in the sentence is in accordance with the data in the table and its meta information. Even a single error should lead to you answering "No". When making the judgment you may be lenient if a number is off by a bit (e.g., reporting 3% instead of 3.5% or "two thirds" instead of 70%), but you should select "no" if any number significantly deviates from the corresponding number in the table.

Inferences based on numbers are okay here, as long as they are not attributed to anyone. For example, "the result merits further study" is acceptable, but "Two men claimed that the results merit further study" is not.

An exception from this question are references to figures - If everything about a sentence is okay,

but it includes a reference to a figure (e.g., "Figure 2 shows X"), you may still mark this question as "yes".

*If you select "No", you do not have to answer Q3 and Q4.*

**(Q3) How many cells does the text cover?** This question asks you to count how many cells a table talks about. You should count as a mention every time a description uses a value from the row or column header, or mentions a table entry. Only cells of the table (anything under START OF TABLE) should be counted. If a description compares multiple values, count all of them even if they are not explicitly mentioned. For example, "X has the highest Y" should count all of the cells that mention "Y" (a statement does not have to be true for the cells to count). The unit of measure and title should not count toward this number.

**(Q4) Generating the text requires reasoning or comparison of multiple cells.** The next question asks whether a sentence requires reasoning. A positive example could compare values in a column or row, e.g., "X has the highest Y", or "X has more Y than Z". If a sentence contains any statement that requires such comparison or implicit reasoning, answer "Yes". In all other cases, answer "No".

### J.3 Example

Below, we show one exemplary table (Tab. 6 with descriptions and corresponding answers.

**Description 1** *Fertility is lowest in Nairobi province (2.8 children per woman), followed by Central province at 3.4 children per woman, and highest in North Eastern province (5.9 children per woman).*
**Q1:** This text is understandable, and the answer is thus "Yes".
**Q2:** All of the details (province names and fertility rates) can be found in the table. The answer is "Yes".
**Q3:** Writing this sentence requires looking at all 8 values of the entire "Province" section in the table, in addition to its section title and the 3 province names. The answer is thus 12.
**Q4:** The text describes "highest" and "lowest" which requires comparison. The answer is thus "Yes" again.

**Description 2** *This is particularly clear in the median ages at which events take place and the*

| Title | Total Fertility Rates by Background Characteristics |
|---|---|
| Unit of Measure | Total fertility rate |
| Kenya | 4.6 |
| Residence | |
| Urban | 2.9 |
| Rural | 5.2 |
| Province | |
| Nairobi | 2.8 |
| Central | 3.4 |
| Coast | 4.8 |
| Eastern | 4.4 |
| Nyanza | 5.4 |
| Rift Valley | 4.7 |
| Western | 5.6 |
| North Eastern | 5.9 |
| Education | |
| No education | 6.7 |
| Primary incomplete | 5.5 |
| Primary complete | 4.9 |
| Secondary+ | 3.1 |

Table 6: The example provided in the annotation instructions.

*compactness of the typical family formation process in each country.*
**Q1:** It is completely unclear what "This" in the sentence refers to and the answer should thus be "No".
**Q2:** Because Q1 is "No", we leave this blank.
**Q3:** Because Q1 is "No", we leave this blank.
**Q4:** Because Q1 is "No", we leave this blank.

**Description 3** *These differentials in fertility are closely associated with disparities in educational levels and knowledge and use of family planning methods*
**Q1:** This sentence is understandable and the answer is thus "Yes".
**Q2:** The mentions of disparities as reasons for the differentials is not attributable to information in the table and the answer should thus be "No".
**Q3:** Because Q2 is "No", we leave this blank.
**Q4:** Because Q2 is "No", we leave this blank.

| Train Setup | Model | STATA QE | STATA REF | STATA QE+REF |
|---|---|---|---|---|
| ar | SSA | -0.00 | 0.45 | 0.22 |
| | Small | -0.04 | 0.36 | 0.16 |
| | XXL | 0.20 | 0.56 | 0.39 |
| en | SSA | -0.01 | 0.48 | 0.24 |
| | Small | 0.01 | 0.53 | 0.29 |
| | XXL | 0.28 | **0.68** | **0.54** |
| fr | SSA | 0.03 | 0.51 | 0.27 |
| | Small | 0.05 | 0.53 | 0.30 |
| | XXL | 0.33 | 0.63 | 0.52 |
| ha | SSA | -0.01 | 0.50 | 0.22 |
| | Small | 0.01 | 0.53 | 0.23 |
| | XXL | 0.20 | 0.55 | 0.32 |
| ig | SSA | 0.09 | 0.53 | 0.25 |
| | Small | 0.03 | 0.44 | 0.21 |
| | XXL | 0.31 | 0.61 | 0.42 |
| pt | SSA | -0.01 | 0.48 | 0.23 |
| | Small | 0.01 | 0.50 | 0.26 |
| | XXL | 0.23 | 0.62 | 0.44 |
| sw | SSA | 0.11 | 0.55 | 0.29 |
| | Small | 0.12 | 0.60 | 0.31 |
| | XXL | **0.43** | **0.72** | **0.55** |
| yo | SSA | 0.11 | 0.49 | 0.26 |
| | Small | 0.08 | 0.44 | 0.24 |
| | XXL | 0.23 | 0.56 | 0.35 |
| MONOLINGUAL | SSA | 0.04 | 0.50 | 0.25 |
| | Small | 0.03 | 0.49 | 0.25 |
| | XXL | 0.28 | 0.62 | 0.44 |
| SKIP NO REFERENCES | SSA | 0.05 | 0.51 | 0.26 |
| | Small | 0.01 | 0.47 | 0.23 |
| | XXL | **0.61** | **0.76** | **0.74** |
| TAGGED | SSA | 0.09 | 0.54 | 0.29 |
| | Small | 0.07 | 0.52 | 0.27 |
| | XXL | 0.57 | **0.76** | 0.69 |
| + SKIP NO VALUES | SSA | 0.11 | 0.57 | 0.32 |
| | Small | 0.10 | 0.54 | 0.30 |
| | XXL | 0.55 | **0.76** | 0.71 |
| + SKIP NO OVERLAP | SSA | 0.01 | 0.47 | 0.22 |
| | Small | 0.00 | 0.42 | 0.19 |
| | XXL | **0.59** | **0.77** | **0.75** |

Table 7: Full evaluation results using STATA. On top, the individual monolingual results, and on the bottom the aggregated results. We highlight notable numbers in the separate sections. In the monolingual setups, Swahili and English lead to the highest performance. The aggregated setups lead on average to a much better performance, with the SKIP NO OVERLAP and SKIP NO REFERENCES setups outperforming the others.

| Language | Model | Understandable | Attributable | Reasoning | Cells |
|---|---|---|---|---|---|
| ar | mT5$_{small}$ | 0.11 | 0.05 | 1.00 | 2.00 |
| | mT5$_{SSA}$ | 0.27 | 0.10 | 1.00 | 4.40 |
| | mT5$_{XXL}$ | 0.78 | 0.50 | 0.86 | 6.05 |
| | reference | 0.86 | 0.56 | 0.84 | 6.11 |
| en | mT5$_{small}$ | 0.28 | 0.05 | 1.00 | 4.67 |
| | mT5$_{SSA}$ | 0.30 | 0.05 | 1.00 | 4.67 |
| | mT5$_{XXL}$ | 0.87 | 0.47 | 0.74 | 6.79 |
| | reference | 0.95 | 0.56 | 0.76 | 7.14 |
| fr | mT5$_{small}$ | 0.14 | 0.22 | 0.33 | 8.17 |
| | mT5$_{SSA}$ | 0.21 | 0.07 | 0.33 | 4.67 |
| | mT5$_{XXL}$ | 0.84 | 0.66 | 0.30 | 5.97 |
| | reference | 0.95 | 0.60 | 0.27 | 7.88 |
| ha | mT5$_{small}$ | 0.20 | 0.11 | 0.50 | 11.00 |
| | mT5$_{SSA}$ | 0.24 | 0.04 | 0.50 | 8.00 |
| | mT5$_{XXL}$ | 0.66 | 0.42 | 0.46 | 8.23 |
| | reference | 0.78 | 0.52 | 0.58 | 12.30 |
| ig | mT5$_{small}$ | 0.10 | 0.59 | 0.79 | 6.07 |
| | mT5$_{SSA}$ | 0.13 | 0.50 | 0.93 | 10.07 |
| | mT5$_{XXL}$ | 0.60 | 0.91 | 0.97 | 8.77 |
| | reference | 0.65 | 0.95 | 0.93 | 9.22 |
| pt | mT5$_{small}$ | 0.14 | 0.08 | 1.00 | 9.50 |
| | mT5$_{SSA}$ | 0.20 | 0.03 | 1.00 | 10.00 |
| | mT5$_{XXL}$ | 0.58 | 0.53 | 0.88 | 8.93 |
| | reference | 0.70 | 0.54 | 0.79 | 7.53 |
| sw | mT5$_{small}$ | 0.23 | 0.48 | 0.86 | 6.29 |
| | mT5$_{SSA}$ | 0.40 | 0.38 | 0.68 | 5.64 |
| | mT5$_{XXL}$ | 0.83 | 0.79 | 0.91 | 9.97 |
| | reference | 0.91 | 0.76 | 0.78 | 8.42 |
| yo | mT5$_{small}$ | 0.21 | 0.10 | 1.00 | 8.25 |
| | mT5$_{SSA}$ | 0.32 | 0.02 | 1.00 | 13.00 |
| | mT5$_{XXL}$ | 0.76 | 0.47 | 0.99 | 8.13 |
| | reference | 0.97 | 0.55 | 1.00 | 5.68 |

Table 8: Full human evaluation results. Note that the attributable fraction is only of those examples marked understandable and reasoning+cells is only answered if an example is attributable.