# OpenReview forum: "TaTA: A Multilingual Table-to-Text Dataset for African Languages"
_EMNLP/2023/Conference — EMNLP 2023 Findings_

### Official Review · Reviewer_KJRS · 2023-08-04

**Typos Grammar Style And Presentation Improvements:** 1. The name of the papers mentions on…
**Soundness:** 3

**Excitement:**

3: Ambivalent: It has merits (e.g., it reports state-of-the-art results, the idea is nice), but there are key weaknesses (e.g., it describes incremental work), and it can significantly benefit from another round of revision. However, I won't object to accepting it if my co-reviewers champion it.

**Paper Topic And Main Contributions:**

The paper present a novel dataset for table-to-text generation for languages spoken in Africa. The authors provide dataset collection description as well as both automatic and human evaluations of models fine-tuned on their data.

**Questions For The Authors:**

Question A: how the task input look like? Is it an image of the graphic or text?

**Reasons To Accept:**

- The proposed dataset is novel;
- The experiments are thoroughly documented.

**Reasons To Reject:**

- The task definition is unclear;
- There is no information about annotators that participated in data collection nor the annotators agreement on the data.
- There are some flows in the presentation.

See presentation improvements section.

**Reproducibility:**

2: Would be hard pressed to reproduce the results. The contribution depends on data that are simply not available outside the author's institution or consortium; not enough details are provided.

**Reviewer Confidence:**

3: Pretty sure, but there's a chance I missed something. Although I have a good feel for this area in general, I did not carefully check the paper's details, e.g., the math, experimental design, or novelty.

---

> ### Author Rebuttal · Authors · 2023-08-28
>
> Thank you for your review and improvement suggestions. We will clarify our task definition and add information about the annotators. Regarding the agreement numbers, we refer to our answer to Reviewer RmMM and lines 367-376 in the paper which state that we worked with the annotators until they reached F1 scores of about 0.9 on an internal test set.
>
> > Question / Suggestion 2: how does the task input look like? Is it an image of the graphic or text?
>
> For the text-based models, the input is provided in Figure 2 and the process of arriving at this representation is described in 3.4. We did not consider image-to-text models as part of this paper.
>
> > Suggestion 1: Clarify notion of “African” and “spoken in Africa” languages and justify Russian.
>
> Thank you for this important point. Outside the main acronym, we tried to avoid referring to the European languages as African languages and only as “languages spoken in Africa” but will work on further disambiguating this confusion. Russian was specifically chosen as a language for which data was available and which is not directly related to any of the other languages to test “real” zero-shot performance.
>
> > Suggestion 3: Table 4 is never references in the main text.
>
> It is referenced at the beginning of Section 5.3 which describes the results in detail.
>
> > Suggestion 4: The information about annotators, their spoken language, the annotators overlap and their agreement should be
> reported.
>
> We will add information about annotators to the paper.

---

### Official Review · Reviewer_qzSY · 2023-08-05

**Typos Grammar Style And Presentation Improvements:** N/A
**Soundness:** 4

**Excitement:**

4: Strong: This paper deepens the understanding of some phenomenon or lowers the barriers to an existing research direction.

**Missing References:**

N/A

**Paper Topic And Main Contributions:**

This paper provides a new dataset for generating text from tabular data in 8 languages, including 7 languages used in Africa (plus Russian). The authors also provide an evaluation of the dataset and of the texts generated. This evaluation shows that half of the texts are not understandable or attributable. They also provide new metrics for human evaluation.

I am unable to evaluate the contribution made by the paper regarding the evaluation metrics, but am highly confident that the dataset and the process of text generation are of great value to the field.


**Questions For The Authors:**

Please explain earlier in the text what "attributable language" means.

Footnote 3 “We use the terms transcribe and transcription as a shorthand for the process where the images of charts and diagrams (info-graphics) and their descriptions are manually converted by human annotators into spreadsheet tabular representations.” Seems to be in contradiction with the statement (page 2) that “Our approach aims to mitigate these issues by transcribing naturally occurring descriptions.” This needs clarification.

**Reasons To Accept:**

Under-documented and under-resourced languages should be a priority of the field, and this paper is a valuable contribution.

**Reasons To Reject:**

None.

**Reproducibility:**

4: Could mostly reproduce the results, but there may be some variation because of sample variance or minor variations in their interpretation of the protocol or method.

**Reviewer Confidence:**

1: Not my area, or paper was hard for me to understand. My evaluation is just an educated guess.

---

> ### Author Rebuttal · Authors · 2023-08-28
>
> Thank you for your kind words. We are excited about contributing to making under-documented and under-resourced languages a higher priority in the *CL community.
>
> In our edits we will move the definition of attribution earlier. We will also clarify that the “issues” in the statement on page 2 refer to crowd workers actively coming up with textual descriptions, rather than just crowd sourcing the task of extracting existing naturally occurring text from PDFs.

---

### Official Review · Reviewer_RmMM · 2023-08-05

**Soundness:** 3

**Excitement:**

3: Ambivalent: It has merits (e.g., it reports state-of-the-art results, the idea is nice), but there are key weaknesses (e.g., it describes incremental work), and it can significantly benefit from another round of revision. However, I won't object to accepting it if my co-reviewers champion it.

**Paper Topic And Main Contributions:**

The authors release a table-to-text dataset consisting of a total of 8,700 instances spread across nine languages, including four African languages. One of the languages, Russian, is added as a test split for zero-shot evaluation. Based on PDF reports spanning a couple of decades on the Demographic and Health Surveys Program by USAID, the authors crowdsourced the transcription and translation for eight languages. Based on the linearized forms of these tables, they perform table-to-text in monolingual, cross-lingual, and multilingual setups (using a common multilingual model). They evaluate the models using a number of evaluation metrics, which consider reference free as well as with reference metrics. Further, propose their own dataset-specific metric, which they demonstrate to correlate better with human ratings than other metrics.

**Questions For The Authors:**

1. I believe the "Reasons To Reject" 2 is a consequence of 1
It would be helpful if the authors can clarify the issues mentioned in "Reasons To Reject" 2 during the rebuttal.

a)  it is mentioned that the test splits have at least three references (line:234), but are these references per language or is it cumulative?

b) Why is it that a manual evaluation of all the references (of test and dev splits) is not considered, with an inter-annotator agreement (on the references alone)?

c) What aspects are different in sTATA which bring more insights to the task? It is not entirely clear how can the same be generalised when preparing future datasets (assuming the required dataset specific model building is possible). In other words, why sTATA is required is not well motivated in the writing.

2. Have you considered using a token free pretrained model, such as byT5, as the african languages may not be sufficiently represented in several of the tokenizer baed pretained models. byT5 has shown to generalie well for several languaegs such as Tamil, Sanskrit etc. as compared to mT5.

**Reasons To Accept:**

- The resource will be a valuable addition to the community - It consists of table-to-text generation data for four African languages, in addition to 5 other

- The evaluation is thorough


**Reasons To Reject:**

1. The draft in its current form is not easy to follow, as the information seems to be scattered all over the place. I believe the writing can be substantially simplified which would only improve the overall quality of the work.


2. While the dataset is thoroughly evaluated for the problem-setting, it is not entirely elaborated on the sanity checks for the human evaluation for the dataset preparation. For instance, it is mentioned that the test splits have at least three references (line:234), but it is not clear if these references are per language or cumulative. Further, it is mentioned that only one reference is manually evaluated (line:359). However, it is not mentioned why that was the case. Especially, since evaluation (and metrics) is a central aspect of the work, I believe more details or justification needs to be provided here. Further, It is not entirely clear to me why the inter-annotator agreement (IAA) wasn't provided for the human evaluation of the test set (or for that matter line 367 to 369 was not entirely clear to me regarding output evaluation and IAA). Similarly, it is stated (and established) that the dataset-specific metric correlated better with human judgment. But it is not entirely clear what aspect is unique to the dataset, where the specific metric is better suited for. In other words, there is nothing stated on how the metric can be generalized in other table-to-text problem settings ( say for future datasets)

3. No details of translators and transcribers are present, including the number of translators.

**Reproducibility:**

3: Could reproduce the results with some difficulty. The settings of parameters are underspecified or subjectively determined; the training/evaluation data are not widely available.

**Reviewer Confidence:**

3: Pretty sure, but there's a chance I missed something. Although I have a good feel for this area in general, I did not carefully check the paper's details, e.g., the math, experimental design, or novelty.

---

> ### Author Rebuttal · Authors · 2023-08-28
>
> Thank you for your thorough reading and posing the list of questions. We will restructure the paper to clarify the dataset configuration and evaluation setups and answer your questions below:
>
> 1a) Since the dataset is fully parallel, the minimum three test reference refer to the number of examples per language. So, for example, if you sample a random English data point, it will have at least three references. Every other language for the same example will have the same number of references.
>
> 1b) We agree that a full human evaluation of every test reference would have been more complete, especially with multiple annotators. Due to budgetary constraints, we had to balance between the following parameters:
>
> - number of evaluated examples
> - number of evaluated languages
> - annotation of dev (yes/no)
> - number of replicated examples
> - number of annotated system outputs
>
> Maximizing all of these would have led to costs higher than constructing the full dataset. For that reason, we chose to maximize the number of evaluated examples, languages, and systems, which came at the cost of not being able to report inter-annotator agreement on the evaluations. To combat this limitation, we compared annotations to a small set of gold-annotations produced by the dataset authors as a qualification test, as reported in lines 367-376.
>
> 1c) Why is StATA required: As shown in Table 3 and discussed in 5.2, no automatic metric has a high correlation with attribution, which means that it is impossible to detect whether a model faithfully describes the underlying table -- the core of the machine learning task. Moreover, in Appendix G we show that standard metrics actually lead to misleading and wrong results when conducting cross-lingual experiments. Thus, for TaTA to be a useful artifact, better metrics are needed.
>
> Is StATA generalizable? While there is some evidence that trained metrics generalize (see, e.g., BLEURT/BERTScore’s ubiquitous application beyond MT), we do not want to make such claim without empirical evidence. The main lesson we learned from our experiments is that (1) metrics need to always be validated in how they are used, and that (2), domain-specific metrics may be necessary. Especially as the field more and more relies on trained metrics (reward models for RLHF, for example), we consider this an important data point to contribute to the discourse and will make this more clear in our paper.
>
> 2) Thank you for the suggestion of using token-free models. We considered ByT5 and for a similar reason experimented with the SSA-T5 model. Since that model did not yield promising results, we did not add additional experiments with ByT5, especially considering that the core contribution of our work is the dataset itself and not the models. Another consideration against ByT5 is that using it for generation is often prohibitively slow, since multiple inference steps are required for every single character.
>
> We will further add more information about the translators and annotators to the paper.

---

### Meta-Review · Area_Chair_nqXv · 2023-09-18

**Recommendation:** 3

**Metareview:**

All the reviewers agree that the paper contributes an extremely valuable multilingual dataset resource for the community as the dataset covers many low-resource languages and is curated carefully via transcription and translation with professional translators. They perform sufficient benchmark experiments and propose a better automatic metric system that better correlates with human ratings than the existing ones.

As mentioned by two other reviewers, and noticed by myself, the paper is slightly hard to read due to the authors' choice of presentation of information in the paper. For instance, the authors currently grouped their proposed metrics STATA under the experimental setup when their intention is to point out existing issues with existing automatic evaluation metrics and hence a need for better metrics (as discussed by the authors in their rebuttal). I believe the current paper structure may be one of the reasons why reviewers RmMM and qzSY have a difficult time evaluating the contributions of the proposed metric STATA, although it's much clearer now (to me) after the authors clarify the contributions of STATA in the rebuttal.

Another weakness of the paper is the missing (important) information about the translators and annotators, which should be added to the revised version of the paper. The authors did not give much details about them in the rebuttal. The paper could have been also made stronger by clarifying the choice of Russian as the zero-shot test language (as the authors did in the rebuttal and should add to the revised section 3.3 of the paper).

---

### Decision · Program_Chairs · 2023-10-07

**Decision:**

Accept-Findings

**Comment:**

All the reviewers agree that the paper contributes an extremely valuable multilingual dataset resource for the community as the dataset covers many low-resource languages and is curated carefully via transcription and translation with professional translators. They perform sufficient benchmark experiments and propose a better automatic metric system that better correlates with human ratings than the existing ones.

As mentioned by two other reviewers, and noticed by myself, the paper is slightly hard to read due to the authors' choice of presentation of information in the paper. For instance, the authors currently grouped their proposed metrics STATA under the experimental setup when their intention is to point out existing issues with existing automatic evaluation metrics and hence a need for better metrics (as discussed by the authors in their rebuttal). I believe the current paper structure may be one of the reasons why reviewers RmMM and qzSY have a difficult time evaluating the contributions of the proposed metric STATA, although it's much clearer now (to me) after the authors clarify the contributions of STATA in the rebuttal.

Another weakness of the paper is the missing (important) information about the translators and annotators, which should be added to the revised version of the paper. The authors did not give much details about them in the rebuttal. The paper could have been also made stronger by clarifying the choice of Russian as the zero-shot test language (as the authors did in the rebuttal and should add to the revised section 3.3 of the paper).